# Attachment, Feeding Practices, Family Routines and Childhood Obesity: A Systematic Review of the Literature

**DOI:** 10.3390/ijerph20085496

**Published:** 2023-04-13

**Authors:** Sarah Clément, Susana Tereno

**Affiliations:** Département de Psychologie, Université de Rouen Normandie, CRFDP UR 7475, F-76000 Rouen, France; susana.tereno@univ-rouen.fr

**Keywords:** child attachment, adult attachment, child obesity, feeding practices, family routines, self-regulation, developmental periods

## Abstract

Childhood obesity is considered a major public health problem. To help prevention and intervention programs targeting families with obese children, this paper is aimed at synthesizing multifactorial and transactional data resulting from studies and reviews assessing relational factors between the child and his or her parents and the child’s obesity risk, including the child’s and CG’s attachment quality, parental feeding practices, and family routines. It is also aimed at assessing the mediation of these links by specific self-regulatory capacities across different developmental periods (0–2, 2–8, and 8–18 years old). The Preferred Reporting Items for Systematic reviews and Meta-Analyses (PRISMA) guidelines were applied in the review methodology. Ten papers were analyzed, including seven empirical studies and three reviews proposing etiological models of childhood obesity. The quality of empirical studies was assessed, and a synthetical model of the results was proposed. This literature review showed that the caregiver’s (CG) and the child’s attachment quality, along with controlling or permissive feeding practices, and few family routines are mostly mediated by appetite dysregulation and emotional regulation strategies with the development of child obesity. New research topics are proposed to understand other facets of childhood obesity, as well as how to better prevent and treat it.

## 1. Introduction

Childhood obesity is a chronic disease that is considered a major public health problem [1]. It is defined by excess body fat and is associated with a risk of cardiometabolic diseases [2], and psychological and relational disorders [3,4]. Excess body fat in children can also put them more at risk of becoming obese adults [5]. Childhood obesity is difficult to prevent and treat because of its complex etiologies. Genetic factors, physical activity, sedentary lifestyle, and access to food have been the main topics of etiological studies [6]. Other studies were conducted to understand better the etiology and risk factors of this disease in order to prevent and intervene effectively amongst obese children and their families, while taking into account the complexity of this disease [7]. Researchers in family therapy have demonstrated that the family environment, such as multiple or dyadic parent–child interactions, are significant in the prevention and treatment of pediatric obesity [8,9]. These factors can induce obesogenic dysfunctional eating behaviors in children [10,11]. This recent body of research helped to better understand the development of childhood obesity and why interventions primarily aimed exclusively at changing parental dietary practices fail in the long term [12]. Developmental researchers have operationalized some of these family and relational factors as the quality of the child’s attachment or the parental feeding practices and family routines, assessing their influence on childhood obesity from a unidimensional perspective [13,14].

The first dyadic parent–child relationship factor studied concerns the parent and the child’s attachment quality. Since the child’s attachment sustains the maturation of his/her brain structures that are involved in the development of self-regulation skills [15,16], it seems to be implicated in the risk of development of childhood overweight and obesity [17,18,19]. More precisely, the caregiver’s (CG) capacity to answer sensibly to the child’s attachment needs conditions the child’s attachment quality developing during the first year. According to attachment theory, the adult’s availability and responsiveness to their children distress cues is influenced by the representations of their own attachment relationships with their CG during childhood [20]. Secure CG are be more capable of perceiving accurately their children’s distress cues and responding to them effectively [20], thereby permitting the child to experience secure relationships in times of distress. In a feeding context, secure parents should be sensible enough to identify and adapt to their children’s feeding, hunger, and satiety cues, and also have a greater emotional attunement, leading to positive and enjoyable feeding sessions [21]. These specific attachment relational patterns between the child and the CG in feeding and non-feeding contexts also condition how the child regulates his/her stress and negative emotions [20,22]. For instance, insecure avoidant and ambivalent children, as well as disorganized children, show more dysregulated stress and negative emotion responses compared with secure children [23,24]. Such dysregulated responses could affect the development of some of their physiological systems such as food intake, and thus influence their weight [25,26].

The second dyadic parent–child relationship factor concerns parental feeding practices. Feeding is a primary parental task during the child’s first year of life. It is also a relevant context to assess the quality of the parent–child relationship [27], which can modulate feeding practices that will influence the development of the child’s eating behaviors [28,29]. In contrast to parenting styles, which are considered as emotional climates between the parent and child and are defined through different levels of warmth/responsiveness and control/demandingness, parental feeding practices are specific behaviors or actions focused on eating, performed intentionally or unintentionally, and used for educational purposes that affect the child’s beliefs, behaviors, and attitudes towards feeding [30]. According to Vaughn et al. they can be divided into three categories [31]: Coercive control refers to practices in which the parent dominates the child so that he/she behaves as the parent desires. It includes restrictive, pressuring, or instrumental and emotional feeding behaviors which meet the parents’ needs more than the child’s needs, including satiety cues. Structuring practices refer to how parents organize the child’s eating environment to promote the consumption of healthy food. They include modeling eating behaviors and repeated exposure to healthy and various foods. Finally, autonomy support practices are practices that enhance the child’s independence and autonomy to help him/her make healthy choices for him or herself. They include encouraging behaviors and non-food rewards. Autonomous and structuring practices are supportive parenting approaches in guiding children to eat healthily while meeting their emotional and physiological needs. These practices would, therefore, have a favorable impact on the child’s weight development [31].

Family routines are organized around different dimensions of family daily life such as bedtime, activities, and mealtimes, making them predictable for the child thanks to their repetition over time [32]. Routines serve a developmental function as they become more organized over time, and children can have a more active role in them [30]. They are also related to the quality of the parent–child relationship and the child’s weight gain [11,33]. Indeed, sharing a family meal four or more times per week is linked with more fruit and vegetable consumption by the child and less consumption of high-calorie foods, which decreases the risk of childhood obesity [34,35]. In their review, Kininmonth et al. [36] highlight that the greater the child’s accessibility to screens, especially in the bedroom [37,38,39], and to high-calorie foods [40,41], the greater the risk of the child’s overweight and obesity. Thus, routines around different dimensions of the child’s life (e.g., meals, screen time etc.) structure the family environment that guides his/her behavior. They also include the family’s emotional climate that supports the child’s development [42], and may become, under certain conditions, a risk factor for child’s obesity development.

Despite the existence of individual links between these factors and childhood obesity, researchers suggest that there are no direct causes of childhood obesity, but rather a transactional process that connects family, interpersonal, and biological systems together [32]. However, the possible mechanisms behind these links had not been assessed. To address this gap, multifactorial and transactional models were recently proposed and tested, demonstrating the mutual influence these factors have on one another, and on the development of childhood obesity [12,43,44]. In addition, the child’s compromised self-regulatory abilities were proposed as a possible mediating mechanism between family routines, feeding practices, the quality of the child’s attachment and the risk of childhood obesity [7,15]. In the literature we found different types of self-regulation: The associations between this illness and general self-regulation ability was mostly inconsistent [45], but statistically stronger with behavioral [46], emotional [47,48], and appetite self-regulation [17] abilities. Therefore, since this is a recent field of research, there is as yet no scientific consensus about which type of self-regulation ability is mostly involved between the factors mention above and the risk of childhood obesity.

The first aim of this systematic review is to synthesize the data on the links between the quality of child’s attachment, parental feeding practices, and family routines and the risk of childhood obesity. The second aim is to assess if these links are mediated by specific self-regulatory capacities. This work addresses two gaps in the literature related to childhood obesity development: (1) The analysis of multifactorial and transactional data to provide a better understanding of children’s food choices and weight trajectories during their development [32]; and (2) The analysis of this subject across different developmental periods to understand how the risks of childhood obesity may evolve [32]. To achieve the latter, we organized our data according to different childhood developmental periods, which to our knowledge, has never been undertaken before. Thus, we propose a model synthesizing the data of our review for each of our identified developmental periods. This approach allows the identification in early development of the individual differences that may influence developmental trajectories [49].

## 2. Materials and Methods

This systematic review is based on an integrative approach of data collection that allows the inclusion of results from studies with different methodologies [50], and on the Preferred Reporting Items for Systematic reviews and Meta-Analyses (PRISMA) method. The inclusion criteria for studies were as follows: (1) independent empirical articles or theses written in French or English; (2) studies dealing with attachment and feeding practices or attachment and family routines and the risk of childhood obesity; (3) studies including overweight or obese children (0–18 years); (4) studies performed over the last 12 years (2010–2022); (5) studies with either a longitudinal or a cross-sectional study design and using a quantitative and/or a qualitative methodology; and (6) literature reviews proposing a multifactorial conceptual model of childhood obesity risk with or without a standardized methodology. Our review did not include the analysis of book chapters or non-peer-reviewed articles. The exclusion criteria were as follows: (1) articles focusing primarily on adults, (2) articles focusing on a childhood obesity intervention program, (3) studies not published in French or English, and (4) studies with samples of children with a mental disorder, mental retardation, or physical illness (e.g., autism, eating disorder, etc.). Based on these criteria, we searched during February and March 2022 for relevant articles. We used three electronic databases: SCOPUS (Science Direct), Pubmed, and Semantic Scholar (see Figure 1). The keywords used were in French and English: “Child obesity AND feeding practice AND attachment” OR “Child obesity AND Family Routines AND attachment” OR “Obésité enfant ET Attachement ET routines familiales”. We chose not to include keywords related to child self-regulation because we wanted to focus our research on attachment, feeding practices and family routines. Self-regulatory abilities are considered here as mediating mechanisms between these factors and childhood obesity, and not as main factors. Furthermore, one of the aims of this review is to assess what types of self-regulation are highlighted in these studies or reviews.

With the combination of all the keywords, a total of 1651 articles were found. There were 1219 articles concerning attachment, feeding practices and childhood obesity, with 989 of these in English and 230 in French. There were 432 regarding attachment, family routines and childhood obesity, with of these 405 in English and 27 in French. After reading the titles and their abstracts, 21 eligible publications were identified and analyzed. The inclusion and exclusion criteria were checked by one reviewer (author 1) and then verified by the second reviewer (author 2). A total of 10 articles met our criteria: 7 empirical articles and 3 literature reviews. For the empirical papers, we collected information about the author(s), the year of publication, the country, the sample’s characteristics (age, ethnicity, socioeconomic level, level of education, BMI, and family structure if noted), the design of the study, the variables evaluated and how they were assessed, the methods for assessing the parent and child’s BMI, the main results, and the quality of the studies. For the literature reviews, we collected information about the author(s), year of publication, country, topic of the review, developmental milestones assessed, and key findings. Then we analyzed the results from the selected papers and assigned the data to three main themes: “Attachment, feeding practices, childhood obesity”, “Attachment, family routines, childhood obesity”, and “Attachment, feeding practices, family routines, childhood obesity”. We also grouped the data into three developmental stages (see Table 1). We determined the boundaries of these stages based on data from the literature about significant developmental milestones in relation to the development of the child’s general-self regulation ability and on the age distribution of the children in the empirical studies and as indicated in the models.

The assessment of the quality of our included papers followed the adapted version of the “National Heart, Lung, and Blood Institute’s Quality Assessment Tool for Observational Cohort and Cross-sectional Studies” [54], a tool consisting of 14 items. Since the selected reviews did not follow a systematic review methodology, only the quality of the included empirical studies could be assessed. The criteria scoring was based on the method used by Beckers et al. and Burnett et al. in their literature reviews [55,56]. One criterion was not considered applicable to the included studies, and therefore removed (i.e., “were the outcome assessors blinded to the exposure status of participants”). This item asks whether the outcome assessors of a study know which participants were exposed to a particular experimental condition. This was not the case in the studies included in our review because the same investigators measured both the experimental exposure of participants and assessed the outcomes. Depending on the study design of the empirical studies, the number of criteria applied was also different: for the two longitudinal studies, 13 criteria were considered in assessing their quality (see Table A1 in Appendix A), and 9 were considered for the cross-sectional studies (see Table A2 in Appendix A). The papers were given scores based on their correspondence with the criteria (0 = no correspondence, 1 = yes). There were four key criteria for longitudinal studies and three for cross-sectional studies (0 = no, 0.5 = partially met, 1 = yes). The total score for each study was calculated as the sum of the scores of the items, and the consideration of the individual scores of the 4 key criteria. The quality of all the empirical papers was assessed and scored by one reviewer (author 1), and then verified by the second reviewer (author 2). Any disagreement between the reviewers was discussed until a consensus was reached. The risk of biased assessments was assessed by one reviewer (author 1) following the Critical Appraisal Skills Programme (CASP) checklists for systematic review and cohort studies [57] (see Table A3 and Table A4 in Appendix A). 

## 3. Results

### 3.1. Characteristics of the Analyzed Papers

Our literature review gathers a total of ten articles, including seven empirical studies and three literature reviews that did not follow a standardized methodology. The publication country of the papers is diverse but mainly English-speaking: the United States (6), the United Kingdom and the United States (1), the United Kingdom(1), Poland (1), and Australia (1). The first article of the decade was published in 2014 and the last in 2020. There are three empirical studies and one literature review on “attachment, feeding practices and childhood obesity”: the USA (1), the USA and the United Kingdom (1), the United Kingdom (1), Australia (1). There are three empirical studies (one of which is included in a thesis) and a review of the literature on “attachment, family routines, and childhood obesity”: the United States (3), Poland (1). There is one empirical study and one literature review on “attachment, eating practices, family routines, and childhood obesity risk”: the USA (2). There are no papers in French on this topic. Only two empirical studies were longitudinal, with the other five being cross-sectional. The empirical studies had several measurement tools: self-administered questionnaires (six studies), observational data (two studies), a survey (one study), an index of a time estimate (one study), and a one-item scale (one study). Two studies also used qualitative methods based on semi-structured interviews. Three studies used mixed methods, combining self-administered questionnaires, semi-structured interviews, observational data, a one-item scale, surveys, and a time estimate index. For all the empirical studies included, the total number of participants was 1325, ranging from a minimum of 77 participants to a maximum of 497. The three literature reviews did not follow a standardized method which means that they have more speculative characteristics. The review published by Fiese and Bost [32] proposes a conceptual model on the regulatory and self-regulatory processes that connect different dimensions (biological, self-regulation, family regulation, food environment) involved in increasing or decreasing the risk of childhood obesity. The review published by Saltzman et al. [58] proposes three different developmental pathways related to the development of childhood obesity, and we chose to concentrate on the “risk” developmental pathway. The review published by Bergmeier et al. [59] proposes a conceptual model that focuses on parent–child relationships to understand how their interactions around feeding can affect a child’s weight status.

### 3.2. Study Quality and Risk of Bias Assessment

Based on the assessment, three studies were rated as “Good” (two cross-sectional and one longitudinal study), and four as “Fair” (three cross-sectional and one longitudinal study). The quality scores of the five cross-sectional studies ranged from 6.5 to 5.5, with an average of 6.7. The quality scores of the two longitudinal studies ranged from 11 to 10.5, with an average of 10.75, which indicates an overall good-quality corpus [54]. None of the three cross-sectional studies reported statistical adjustments for potential confounding variables [60,61,62], two did not report inclusion and exclusion criteria [61,63], and one did not present a sample size justification [62]. One longitudinal study did not present inclusion and exclusion criteria or report all statistical adjustment for potential confounding variables [21]. Another study did not state clear times points of measurements [45]. Following the Critical Appraisal Skills Program (CASP) checklists for systematic review and cohort studies [57], all the cross-sectional and longitudinal studies addressed a clearly focused issue, recruited their cohort in an acceptable way, had replicable and comparable results with other evidence, and used validated tools [21,45,60,61,62,63,64], with the exception of one cross-sectional study that used a non-validated tool [61]. The cross-sectional studies had data based on self-reports, which provides a less robust basis for changes in clinical practice [60,61,62,63,64]. The longitudinal studies included either observational data or a mixed methodology, which provide robust evidence for recommendations of change in clinical practice [21,45]. In general, the included studies had limited bias. The three reviews clearly addressed their topic and all the important outcomes were considered [32,58,59]. However, since the authors did not use a standardized methodology, there is no clear information on how they screened the included papers, on their quality, or the replicability of results for a local population. Even if the results of their review are precisely synthesized and important outcomes considered, the lack of information concerning the used methodology, paper screening, sources and quality of included papers indicate potential bias.

### 3.3. Analysis of Papers

#### First Developmental Period (0–2 Years)

The studies reviewed for the 0–2 years developmental period are described in Table 2. In general, insecurely attached young children seem to be at risk of gaining weight via compromised general self-regulation [32,58,59]. The CG and the child’s insecure attachment, in addition to family risk factors, can directly affect the development of the child’s appetite self-regulation abilities, and indirectly via poor parental responsiveness to feeding [58]. Secure fathers are more attuned to their infants during feeding in contrast to dismissing fathers. Fathers with unresolved attachment trauma use more controlling behaviors, which may compromise the development of the child’s eating self-regulation [21]. Parents using permissive and indulgent feeding practices put their child at risk of overweight and obesity through emotional eating and their answer to the child’s negative emotions, these factors being related to their attachment quality [32]. Finally, a higher number of routines around dinner was linked with less appetite dysregulation in children with highly insecure mothers, and conversely, the presence of “Household Chaos (HC)” was associated with higher levels of appetite dysregulation in children whose mothers also reported low levels of emotional responsiveness [45].

### 3.4. Second Developmental Period (2–8 Years)

The studies reviewed for the 2–8 years developmental period are described in Table 3. Globally, insecure CGs seemed to have fewer mealtime routines and allowed their children more screen time, which in turn predicted their children’s consumption of unhealthy foods. They tended to use negative emotional regulation strategies and had emotional pressuring feeding styles that are related to unhealthy food consumption in children [64]. More specifically, anxious attachment CGs seem to more frequently have children with a diminished eating self-regulation ability, with this association being mediated by controlling/persuasive feeding practices [63]. Maternal anxious attachment is also linked with emotional feeding practices and emotional eating in children and pre-adolescents. These mothers used emotional feeding practices primarily in response to the child’s emotional eating [60].

### 3.5. Third Developmental Period (8–18 Years)

The studies reviewed for the 8–18 years developmental period are described in Table 4. To summarize, from middle childhood to teenagehood, children with insecure attachment are more likely to have appetite dysregulation, leading them to consume more high-calorie foods and to engage in obesogenic behaviors. CGs’ insecure attachment is linked with an emotional and social eating regulation: anxious attachment predicts emotional eating, and avoidant attachment poorer control and organization of nutrition [61]. There is a link between the CGs’ obesogenic behaviors and the transmission of such behaviors to their children in general and feeding contexts [62]. Such modeling is influenced by their attachment quality [61]. Yet, the CG’s obesogenic behaviors and their transmission are not linked with the child/adolescent’s own obesogenic behavior. To conclude, appetite or eating dysregulation and emotional regulation strategies are the forms of self-regulation most frequently found in the interactions between our primary factors. General self-regulatory abilities were only mentioned in the three models but were not considered in the empirical studies.

## 4. Discussion

This paper aimed at synthesizing multifactorial and transactional data resulting from studies and reviews assessing the links between the child’s and CG’s attachment quality, parental feeding practices, family routines and the risk of childhood obesity across three developmental periods (see Figure 2). It also aimed to assess the mediation of these links by specific self-regulatory capacities across different developmental periods. In general, this literature review showed that the CG’s and child’s attachment quality was associated with controlling or permissive feeding practices, few family routines, and the modeling of obesogenic behaviors. These were mostly mediated by appetite dysregulation and emotional regulation strategies and influenced the child’s food consumption and weight trajectory toward overweight and obesity status.

### 4.1. First Developmental Period (0–2 Years)

Among the most studied concepts at this developmental period, the quality of parental attachment in the three models presented in the literature review papers [32,58,59] and in the two empirical studies [21,45], as well as the quality of infant attachment presented in the same models, were considered. Other factors affecting the CG–child relational quality were operationalized through feeding and emotional responsiveness in two of the models [58,59] and in one of the studies [45]. Indeed, feeding is a context that contributes to the formation of the early child attachment relationship [65]. Secure parents have more sensible responses to the child’s eating behaviors and cues, which supports the child’s innate ability to self-regulate their food intake [15]. Conversely, less sensible interactions with an insecure CG, or especially with unresolved attachment trauma, during feeding can compromise this self-regulation ability [12], leading to a risk of weight gain. On the one hand, detached fathers in Reisz et al.’s study [21] are less attuned with their child during feeding as a result of an emotional deactivation strategy for coping with a potentially stressful feeding context or to minimize the importance of relationships [20]. On the other hand, when facing the hungry child’s distress, and similar to the results of Messina et al. [66], unresolved trauma memories could reactivate within fathers and lead to an attempt of the CG to regain control using controlling feeding practices [12,21]. Among the least studied concepts during this developmental period, feeding practices were included in two models [32,59] and assessed in one study. Reisz et al. demonstrated that fathers with unresolved attachment trauma had more controlling feeding practices with their sons than with their daughters [21]. These findings follow the results of studies in which fathers are reported to engage in more controlling practices in general than mothers [8]. Using this type of practice, CGs perceive less well the child’s signals and cues [30], which prevents the child from learning to identify and correctly regulate his or her physiological signals of hunger/satiety [67]. This may increase the risk of emerging difficulties in his or her eating self-regulation skills [68], and so of weight gain. Finally, family routines were included in two models [32,58] and in one study [45]. In their “at-risk” model, Saltzman et al. proposed that in addition to poor attachment quality and the CG’s low responsiveness to eating, some family routines would affect the development of the child’s self-regulatory abilities [58]. This hypothesis was confirmed by the same author [45]. Routines can be important for children exposed to frequent stress, including an insecure attachment relationship, because they provide stable and predictable interactions [45]. When routines are disrupted or unstable, the child’s environment can become chaotic [69], which prevents the fluidity of interactions essential for a healthy development and underlies the links between this type of family environment and childhood obesity. 

### 4.2. Second Developmental Period (2–8 Years)

Parental attachment, which is among the most studied concepts in this developmental period, was assessed in four of the studies [45,60,63,64], with the CG’s anxious attachment mainly emphasized in these data. Feeding practices were also frequently assessed in this review and included in three studies [60,63,64]. The CG’s anxious attachment was indirectly linked with the child’s eating self-regulation ability through persuasive controlling feeding practices [63]. This kind of feeding practice is the only one that has been longitudinally related to child overweight [56]. Because anxious CGs have little ability to manage their distress, they may activate their own attachment system when facing their child’s distress due to hunger. Therefore, they can use controlling feeding practices in response to their anxiety [66]. As Hardman et al. demonstrated, they may be at risk of teaching their child dysfunctional emotional regulation strategies such as the use of food to regulate negative emotions [60]. If used consistently, and through a parent–child transmission mechanism, these strategies seem to induce emotional eating within the child [61,70,71]. Indeed, anxiously attached CGs have lower distress regulation capacities. Relying on external sources to help them to manage food consumption puts them at risk of developing emotional eating [72]. Thus, we hypothesize that the child seems to integrate the CG’s regulation model at early stages of development, which induces a misidentification/confusion of his/her emotional signals with hunger/satiety physiological signals, and therefore, increases emotional eating. As demonstrated by Hardman et al., anxious mothers tend to reinforce this regulation strategy by responding to their child’s emotional overeating with more emotional eating strategies that are considered controlling feeding practices [60]. This is an interesting finding that is consistent with previous research indicating that maternal feeding practice is firstly “child responsive” [70] and that the child’s overweight status is not primally induced by a controlling feeding practice as was often stated before [73,74,75]. Family routines around meals and other dimensions of family life such as screen time were only assessed in two studies. Bost et al. reported that insecure parents tended to have fewer mealtime routines and more television time [64], which is a risk factor for childhood obesity [11]. More screen time could be interpreted as a way to reject or to avoid interactions and the child’s negative emotions [64]. 

### 4.3. Third Developmental Period (8–18 Years)

One of the concepts focused on in both studies [61,62] that investigate this period are the CG’s and child’s “obesogenic behavior”, including mealtime routines and their “modeling” (conceptually similar to feeding practices) towards the child or adolescent. There were no associations between the child/adolescent obesogenic behavior and the CG’s own obesogenic behavior and modeling [62], which can be explained by the fact that children who are 8–18 years old spend less time with their CGs, and therefore, are less influenced by their family environment. Thus, the influence of parental obesogenic behavior modeling on the child’s own behaviors may be strongest when the child is young and predominantly heteroregulated. Concerning child attachment quality, it was only assessed in one of the studies [62] which demonstrated that insecure children/teenagers had a higher risk of consuming high-calorie foods. Interestingly, and in contrast to the adult sample in this review, it was the avoidant children who had more “obesogenic” behaviors, this link being mediated by their eating self-regulation ability. The latter was operationalized by emotional eating and low behavioral regulation, and given that emotional eating is also considered as an emotional avoidance strategy [76], we hypothesize that the use of eating to manage emotions could be part of an emotional suppression strategy found in this type of attachment. An avoidant attachment, and the use of emotional eating as an avoidance strategy, could impoverish the child’s emotional awareness [77] and lead to the risk of developing alexithymia, a psychopathology linked with avoidant teenager profiles in the literature [78], and increased risk of overweight and obesity [79]. Future studies should test the association between the child/teenager’s avoidant attachment, the regular recourse to emotional eating as an emotional regulator, the development of alexithymia, and weight gain. 

### 4.4. Practical Applications

From infancy till middle childhood, children are highly influenced by their family environment and their interactions with their CG [51]. When facing families in which children are overweight or obese, presenting along with their CG’s insecure or unresolved attachment cues, the latter having also controlling feeding practices and few family routines, clinicians can target these relational factors to help these families achieve healthier habits, behaviors, emotional and appetite regulation strategies. Assessing the level of motivation of the CGs to change their routines and feeding practices seems important since changing habits can be difficult in the longer term [80]. Clinicians can use motivational interview techniques [81] to help them identify unhealthy routines. Stable routines are important protective factors for children with insecure mothers [45], making them key elements to investigate for clinicians. They can also assess the type of feeding practices CG’s use and work on to establish more stable and predictable routines and to introduce more structure and autonomy in feeding practices. Clinicians need to assess the quality of the CG’s attachment representations in order to see how they influence the interactions between the adult and the child during stressful events (i.e., when a child cries during feeding). Since unresolved trauma attachment memories and insecure attachment representations influence the CG’s interpretation of the child’s emotional cues and their behavior [12,21], practicians can help these CG’s to better understand their attachment needs and those of their child to identify and respond to them more sensibly. In a feeding context, this will help the CG discriminate between attachment/emotional cues and hunger/satiety cues expressed by the child, which will also help the child in learning how to better regulate him or herself within these dimensions. Practicians can assess the CG’s and child’s attachment quality, as well as how the CG manages the child’s feeding and emotional needs, by observing their interactions during a meal in ecological or clinical settings. For attachment quality more specifically, practicians can use several tools including the Massie–Campbell Mother–Infant Attachment Indicator During Stress Scale (0–18 months) [82], which is a standardized observation of attachment behaviors in small stress context that encompasses several family routines such as dressing, bathing, and feeding contexts, and reunion episodes. An additional tool is the Coding Interactive Behavior system which helps to code interactions between the child (between 2 and 36 months) and his or her CG to assess maternal sensibility [83]. In our corpus, maternal anxious attachment was linked with emotional eating within children and pre-adolescents [60]. If the CG or children present emotional eating, practicians can work with them on developing other emotional regulation skills and help patients to discriminate more easily emotional and hunger/satiety clues, thereby reducing confusion. Dialectical behavioral therapy (DBT) [84] or acceptance and commitment therapy (ACT) [85] are therapeutic interventions that have proven to help adults or teenagers with emotional eating and overweight/obesity [86]. The Dutch Eating Behavior Questionnaire (DEBQ) is a widely used tool that helps to identify emotional eating [87]. When children are old enough to be able to regulate themselves, they are less influenced by the obesogenic routines and behaviors of their family members [62]. Since insecure avoidant teenagers have a higher risk of consuming high-calorie foods [62], practicians need to work on the quality of the therapeutic relationship before engaging in a specific therapeutic or motivational approach so that the relational experience modulates the patient’s internal working models concerning attachment needs. Diverse therapeutic approaches can be proposed including ACT and DBT, but also Fonagy’s and Bateman’s mentalisation-based treatment [88] that helps patients access their internal experiences and consider differently their thoughts and emotions [89]. As previously mentioned, assessing their daily routines and feeding habits can be useful to identify obesogenic behaviors and work on their motivation to change such behaviors.

### 4.5. Study Limitations and Future Research Directions

The analyzed studies show important findings and reflections on our subject, but there are also some limitations that raise several future research directions. Concerning attachment quality, of the 10 studies reviewed, only Lamson et al. assessed the child or adolescent’s attachment [62]. Given that children’s attachment is constructed during the first year of life, it would seem important to consider this variable in future studies of infants and preschoolers. Additionally, only the study by Reisz et al. addressed unresolved representations of attachment in adults [21]; the other six studies focused on the axes of security or avoidant/anxious insecurity. It also appears important to consider parental mental health since it can interfere with the CG’s ability to respond consistently and sensitively to the child’s needs [59] and some of the child’s temperament dimensions [32,59] associated with dysfunctional eating behaviors [59]. Moreover, when obesogenic behaviors are assessed, they are often reduced to food consumption, whereas Lamson et al. also included mealtime routines and other dimensions of family life [62]. Socio-economic level was not identified in the studies of Hardman et al. or Lamson et al. [60,62]. Education level was not assessed in the study of Powell et al., and child’s gender was not considered in the studies of Hardman et al. and Pasztak-Opilka et al. Ethnicity, socio-economic and educational level, and gender are important demographic factors to control in the data analysis as potentially affecting the development of the child’s weight status [21,90,91]. The studies by Powell et al., Hardman et al., Pasztak-Opilka et al., Bost et al., and Lamson et al. used only self-report questionnaires, which may be subject to auto-perception bias. The validity of the results should be strengthened by using other means of assessment, so the use of a mixed method is recommended as most of the studies are also cross-sectional. Finally, it is necessary for future studies to include more fathers in their samples.

## 5. Conclusions

This literature review has highlighted the impact of dyadic and multiple family relationship factors on the development of the child’s weight status. Thus, the quality of parental and child attachment, parental feeding practices and family routines can induce dysfunctional eating behaviors and compromise the child’s self-regulation capacities, especially the child’s eating and emotional self-regulation, which can result in weight gain and child obesity. The results of this review are part of a recent research focus on childhood obesity, and we propose new research topics to understand other facets of this illness, as well as how to better prevent and treat it by changing the child’s environment on several relational levels.

## Figures and Tables

**Figure 1 ijerph-20-05496-f001:**
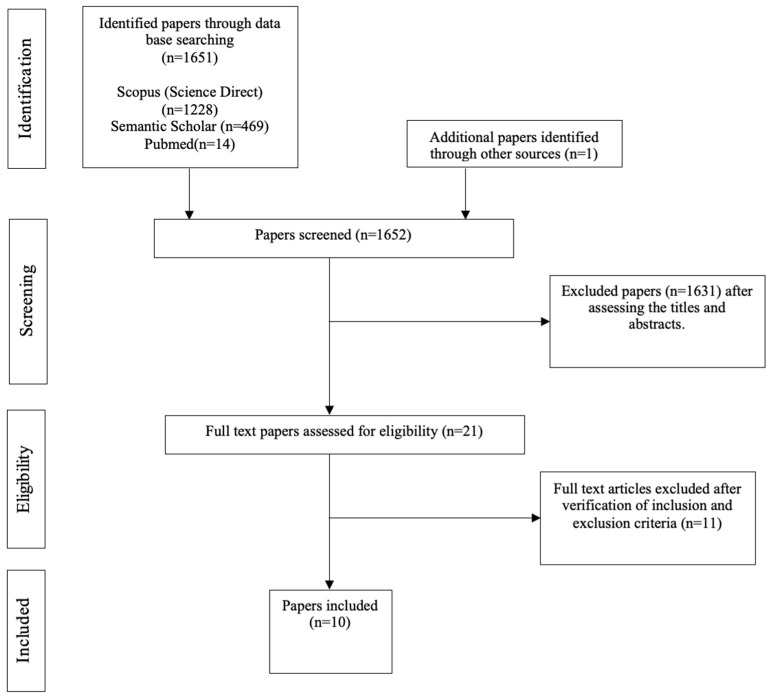
PRISMA flowchart of paper selection research.

**Figure 2 ijerph-20-05496-f002:**
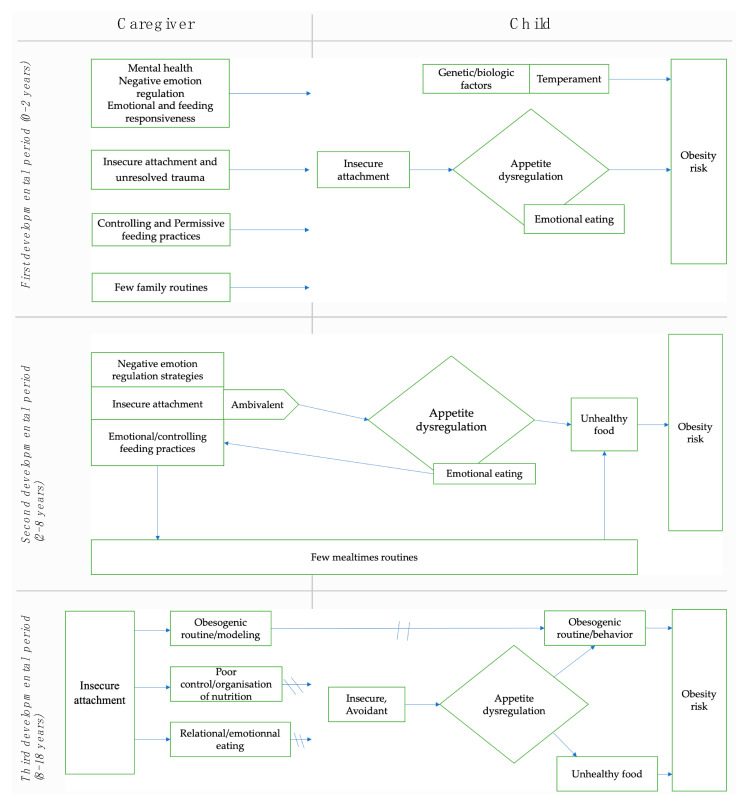
Synthesis of the literature review for the three developmental periods.

**Table 1 ijerph-20-05496-t001:** Developmental periods.

Developmental Period	Characteristics
Early developmental period (heteroregulation); 0–2 years.	The child’s self-regulatory processes are primarily conditioned by his/her environment, particularly by the GC. The latter are directly involved in modulating the child’s expressions at a feeding, emotional, and behavioral level [51].
Second developmental period (transition from heteroregulation to more autonomous regulation); 2–8 years.	The child becomes more autonomous, acquires and applies self-regulation strategies but is still be dependent on interpersonal regulation, especially CG responses. As he/she grows and becomes less dependent on the parent–child dyadic relationship, the child turns to extended social support from friends, and other adults in his/her environment [51,52].
Third developmental period (autonomy); 8–18 years.	The child/adolescent is considered autonomous in his or her self-regulation capacities. He/she is an active partner in the relationship with an individualized opinion [53], no longer needing the proximity with his CG to help him/her self-regulate, but their availability and accessibility [53]. The child has a repertoire of numerous self-regulation strategies provided by the different interpersonal past and present experiences.

**Table 2 ijerph-20-05496-t002:** Studies focused on the first developmental period (0–2 years).

a. Attachment, Feeding Practices, Childhood Obesity			
Reference	Sample	Method	Key Findings
Reisz et al. (2019) [21] Fathers’ attachment representations and infant feeding practices. UK and USA **Quality:** Fair	n = 118 **Fathers** **Mean Age:** 30 year-old (y-o) fathers **Child:** **Age:** From birth to 8-month-old (m-o) infants 41% female **Ethnicity: ** 82% white **Education: ** Well educated **Socio-economic Status (SES): ** Middle-class	**Design:** Empirical, longitudinal (L) (11 months) **Measures:** **T1** (Third trimester of pregnancy) Semi-structured interview Adult Attachment Interview (AAI): Autonomous, dismissing, preoccupied, unresolved state of mind **T2** (8-m-o infants): Observational assessment tool (Feeding Scale: videotape and coding scale): attunement, conflict, control **Child’s weight:** not reported (NR)	Fathers were more controlling with their sons than with their daughters, regardless of their attachment representation (*p* = 0.02). **Secure fathers:** - More attuned with their children during feeding (*p* < 0.05). - Perceive their needs more adaptively. - Respond to them more appropriately. **Dismissing fathers:** - Less attunement with their child during feeding (*p* < 0.05). - Less in conflict or in control of their food consumption. **Preoccupied fathers:** - Insufficient sample size to support the idea that they are more conflicted, in control, and less attuned with their child. **Fathers “unresolved “:** - At greater risk for controlling feeding practices (*p* < 0.01).
**Reference**	**Method**	**Key findings**	
Bergmeier, et al., 2020 [59] Early mother–child dyadic pathways to childhood obesity risk: A conceptual model. Australia **Quality:** Not measured (NM)	**Literature review:** Focus on parent–child relationships to help understand pathways that lead from parent–child feeding interactions to child body mass index **Developmental stages included:** - Infancy to toddlerhood - Preschool years	**Attachment:** - The caregiver’s responses to the child’s emotional states and needs during times of fear/distress help the child regulate his/her physiological and emotional reactions. - Potential risk mechanism: the child’s compromised self-regulation ability due to dysfunctional CG–child interactions. **Temperament:** - Plays a role in establishing the quality of parent–child interactions. - Certain dimensions could be associated with dysfunctional eating behaviors in children. **Mother’s mental health status:** - Implicated in a lower quality of parent–child interactions. - Influences the development of insecure child attachment during the first year. - Interferes the maternal ability to give consistent/sensitive responses and caregiving. **Mutually Responsive Orientation:** - Characterized by mutual parent–child responsiveness and shared positive affect - Emerges in the dyadic feeding relationship when children transit to solids and are being socialized to become independent eaters. - Can reduce the frequency and severity of mealtime conflict. - Can promote the child’s internalization of parent’s food attitudes and behaviors. **Problematic feeding practices:** - Controlling food excessively, missing the child’s cues, expressing more negative emotions and conflicts with the child in food-related situations compared with non-food situations. - Excessive parental control over food is implicated in the development of problematic eating behaviors in children. **Self-regulation:** - Positive parent–child relationships support the development of emotional self-regulation and help establish attachment security. - This promotes an optimal development of neurophysiological systems, particularly those involved in stress response, sleep, appetite, and a long-term ability to self-regulate. - Risk factor for the development of obesity: a low self-regulation ability in food or non-food related contexts.	
**b. Attachment, family routines, and childhood obesity**			
**Reference**	**Sample**	**Method**	**Key Findings**
Saltzman et al. (2018b) [45] Family chaos, attachment security, and responsiveness: Associations with appetite self-regulation in early childhood USA Quality: Good	n = 110 families **Parent Mean Age:** 30.90 years **Maternal** **ethnicity:** 80.90% white **Education:** Well educated **SES:** Average, high income **Child:** **Mean Age:** 20.97 months (17.80–34.90) 51.40% female	**Design:** Empirical, L However, (CD) time points not determined **Measures:** **Family:** Mealtime routines and Household Chaos (HC): **Mother self-report:** CHAOS: Confusion, Hubbub and Order Scale (CHAOS); Family Ritual Questionnaire (FRQ); Relationship Scales Questionnaire (RSQ); Caregiver Feeding Styles Questionnaire (CFSQ); Coping with Children’s Negative Emotions Scale (CCNES); Children’s Eating Behavior Questionnaire (CEBQ); Infant Behaviors Questionnaire -Revised (IBQR). **Maternal attachment:** Scriptedness and insecurity (dismissing) and fearful: Semi-structured interview: The Attachment Script Assessment (ASA) **Responsiveness:** (feeding and emotional responsiveness during mealtimes and in general) and Child’s appetite dysregulation (Responsiveness to food and emotional overeating) **Observational data:** The observational Feeding Behavior Coding System (FBCS method; mealtime videos) **Child’s weight:** NR	HC, feeding routines, and maternal responsiveness are directly associated with the child’s appetite dysregulation (*p* < 0.05). More dinner routines were associated with less appetite dysregulation in children with highly insecure mothers (*p* < 0.05). Dinner routines were not associated with appetite dysregulation in children of more insecure mothers; they are considered protective factors for these children (*p* < 0.05). HC was associated with higher levels of appetite dysregulation in children whose mothers had low levels of emotional responsiveness (*p* < 0.05). HC was not associated with different levels of appetite dysregulation in children whose mothers reported high levels of emotional responsiveness (*p* < 0.05).
**Reference**	**Method**	**Key Findings**	
Saltzman et al. (2018a) [58] Development of appetite self-regulation: Integrating perspectives from attachment and family systems theory USA **Quality:** NR	**Literature review:** Includes research from attachment and family system theories regarding the influence of individual, dyadic, and family factors on the development of children’s appetite self-regulation abilities **Developmental stages included:** Infancy–Early childhood	**Risk pathway:** - Poor attachment quality and family risk factors (HC, poor family routines etc.) independently affect the infant’s self-regulation appetite directly and indirectly via poor responsiveness to feeding. **Resilience pathway:** - Family factors can be protective against other risk factors, such as a poor attachment quality, by promoting, via feeding responsiveness, a better appetite self-regulation within children. - Family functioning can help promote the child’s perception of parental sensibility, even in families exposed to adversity. - Low HC and regular family routines can help promote the child’s sensibility and self-regulation. **Well-being pathway:** - Situations where children can develop an optimal appetite self-regulation. - Parents and children have a secure attachment quality, a high family functioning system, low HC and positive interactions during mealtimes. - Secure attachment and family factors help parents use sensible feeding practices, which predicts optimal self-regulation within children.	
**c. Attachment, feeding practices, family routines, childhood obesity**			
**Reference**	**Method**	**Key Findings**	
Fiese and Bost (2016) [32] Family ecologies and child risk for obesity: Focus on regulatory processes USA **Quality: NR**	**Literature review:** Focus on specific proximal regulatory processes that connect different ecologies (biological regulation, child regulation, family regulation, food environment regulation) implicated in the increase or decrease in child’s obesity risk **Developmental stages included:** CD	**Individual biology** - Unhealthy gut microbiota can have a negative effect on the gut–brain connection, and central nervous system functioning. - Conversely, stress triggered in the central nervous system can affect gut microbiota. - Proximal interactions in family during early childhood influence physiological stress regulation, which in turn, can affect inflammation states. - Obesity is considered as a condition of low-grade chronic inflammation. **Child temperament, self-regulation, and child obesity:** - Negative reactivity and emotion regulation are linked with child weight-related outcomes. - Consistent findings when assessing child temperamental self-regulation and weight outcomes (more than temperamental reactivity). - Relationship between temperament and child BMI may also be moderated by parental feeding practices, maternal sensitivity, and developmental period. **Links between attachment relationships and child obesity:** - Parent–child attachment relationships contribute to emerging emotional and behavioral responses and regulation patterns. - They could have an influence on the interpersonal context through which eating behaviors are socialized. - Attachment quality acts also as a moderating factor, combined with child temperament. **Feeding styles and parenting practices:** - Permissive and indulgent feeding styles do not facilitate the child’s attendance to satiety cues, giving him or her little guidance. - Such a feeding style context does not give the child the opportunity to develop competence in regulating healthy food preferences or age-appropriate portion size - One potential mechanism between feeding styles and child obesity is through emotional eating and response to negative emotions. **Family routines:** Family routines represent a context where the child’s negative emotions can be answered to and socialized, especially during mealtimes. If negative affect is associated with a struggle around eating, there is less opportunity for a positive learning of self-regulation. When the family environment is unpredictable, then the fluidity of essential interactions for a healthy development is disrupted.	

**Table 3 ijerph-20-05496-t003:** Studies focused on the second developmental period (2–8 years).

a. Attachment, Feeding Practices, Childhood Obesity			
Reference	Sample	Method	Key Findings
Powell et al., (2017) [63] The relationship between adult attachment orientation and child self-regulation in eating: The mediating role of persuasive-controlling feeding practices. USA **Quality:** Good	n = 265 68.3% **mothers** Mean Age = 31.37 **Child:** **Mean Age** = 4.17 (range 2.12–7.03) **Ethnicity:** 72.9% White **SES:** Average, high income	**Design:** Empirical, cross-sectional (CS) **Measures:** Self-reports: **Adult attachment: ** Relationship structures questionnaire of the Experiences in Close Relationships—Revised (ECR-RS): anxiety and avoidance **Parental feeding practice:** The Feeding Practices and Structure Questionnaire (FPSQ): persuasive feeding (reward for behavior and for eating) **Child eating self-regulation:** Child Eating and Behavior Questionnaire (CEBQ): self-regulated eating (emotional overeating, food responsiveness) **Child’s weight:** NR	The relationship between anxious parental attachment style and the child’s self-regulatory abilities is significant when mediated by persuasive feeding. **High levels of anxious parental attachment:** - May lead to persuasive controlling feeding practices with the child (*p* < 0.001) and the child’s decreased ability to self-regulate in a feeding context (*p* < 0.001). No significant associations between avoidant parental attachment and controlling feeding practices or the child’s ability to self-regulate feeding. **When controlling for gender, the significant associations were stronger for fathers than for mothers** (*p* < 0.001).
Hardman et al. (2016) [60] Using food to soothe: Maternal attachment anxiety is associated with child emotional eating. UK **Quality:** Fair	n = 77 **Mother Mean Age:** 39.23 **BMI:** 25.93 **Child Mean Age:** 8.3 (range 3–12) 51% **female** **BMI** z-scores: 0.17 **Education:** Majority educated	**Design:** Empirical, CS **Measures**, Self-reports: **Maternal:** **Attachment:** Experiences in close Relationships (ERC): anxiety **Disinhibited eating:** Three Factor Eating Questionnaire (TFEQ) **Emotional feeding strategies:** Parental Feeding Strategies Questionnaire (PFSQ). **Child emotional eating** (CEBQ) **Weight:** Self-report (mother), BMI Z-scores (child)	**Mothers with Anxious attachment:** - Had children with more emotional overeating (*p* < 0.01). - Are more likely to use emotional feeding strategies with their child (*p* = 0.01), which was associated with the child’s increased emotional eating (*p* = 0.004). - **Used emotional feeding strategies primarily in response to the child’s emotional overeating** (*p* = 0.001).
**b. Attachment, feeding practices, family routines, childhood obesity**			
**Reference**	**Sample**	**Method**	**Key findings**
Bost et al. (2014) [64] Associations between adult attachment style, emotion regulation, and preschool children’s food consumption. USA **Quality:** Fair	n = 497 families **Caregiver’s Mean Age:** 32.45 90% **female** **Education** Well educated **SES** Average **Child Mean Age** (month): 39.04 (range: 2.5–3.5 years) **Ethnicity** 70%: White	**Design:** Empirical, CS **Measures:** Self-report: **Adult attachment:** (RSQ): anxious/fearful, dismissing/avoidant, secure **CG responses to their children’s negative emotion:** The Coping with Children’s Negative Emotion Scale (CCNES): distress, punitive, minimization emotion or problem focused reaction Expressive encouragement **CG feeding styles:** Comprehensive Feeding Practices Questionnaire (CFPQ): emotion-related pressuring feeding and healthy eating styles **Family Mealtime Routine:** Family Ritual Questionnaire (FRQ): frequency, planning, communication **Child television viewing and food consumption:** Early Childhood Longitudinal Study-B (ECLS-B): index of total minutes child television viewing **CG depression/anxiety:** Depression Anxiety Stress Scales (DASS) **Weight:** CG BMI communicated in descriptive data, no mention of **child** BMI, nor the assessment weight method	**Links between:** - The CG’s attachment insecurity and negative emotional regulation strategies of the child’s distress (*p* = 0.001). - Emotional feeding styles and unhealthy food consumption (*p* < 0.001). **Insecure parents:** - Had fewer mealtime routines (*p* < 0.01). - Allowed more television time (*p* < 0.001). - Had emotion-related feeding styles more frequently than secure CGs (*p* < 0.05). - Their children consumed more unhealthy foods (salty snacks, sugary drinks, and fast food) (*p* < 0.01). **CGs who responded more negatively to the child’s distress:** - Had fewer family routines (*p* < 0.01). - Were more likely to use emotional feeding styles (*p* < 0.001). - Allowed more television time (*p* < 0.01). **CG’s positive emotional regulation:** - Significantly related to children’s fruit/vegetable consumption (*p* < 0.001) and to modeling a healthy diet (*p* < 0.001).

**Table 4 ijerph-20-05496-t004:** Studies focused on the third developmental period (8–18 years).

a. Attachment, Family Routines, Childhood Obesity			
Reference	Sample	Method	Key Findings
Pasztak-Opiłka et al. (2020) [61] Adult attachment styles and mothers’ life satisfaction in relation to eating behaviors in the families with overweight and obese children Poland **Quality:** Fair	n = 52 dyads **Mothers Mean Age** = 41.81 **Child:** **Mean Age:** 14 (range: 11–18 years) **BMI:** 19% overweight 81% obesity **Education:** Secondary school (42%) **SES:** Average and high income	**Design:** Empirical, CS **Measures:** Self-reports: **Mother’s attachment: ** Questionnaire of Attachment styles (KSP): secure, anxious, avoidant **Mother’s life satisfaction: ** Satisfaction with life scale (SWLS) **Mother’s Eating Behavior (EB): ** Eating behaviors questionnaire (KZZ): Positive EB: knowledge and control of nutrition. Negative EB: negative beliefs and cultural customs; regulation of family relationships and emotions; incorrect organization of nutrition **One-item scale for financial situation.** **Weight and height:** mother’s self-report **Objective measures:** Medical diagnosis based on centile grids	**Insecure attachment:** - Linked with the tendency to regulate emotions and family relationships through eating. **Anxious attachment:** - Predicts emotional eating (*p* < 0.01). **Avoidant attachment:** - Is related to poorer control and organization of feeding (*p* < 0.05). **Mothers with an average weight:** - Had more positive eating behaviors compared with overweight mothers (*p* < 0.05). **Obese mothers:** - Control and regulate their emotions more often through eating (*p* < 0.01). **Maternal BMI:** - Predicts the establishment of negative EBs to regulate emotional states (*p* = NR).
Lamson et al. (2020) [62] Attachment, Parenting, and Obesogenic Behavior: A Dyadic Perspective USA **Quality:** Fair	n = 77 dyads. 86.6% female **Parents Mean Age:** 37.29 years **Education:** Elementary and Middle school (48%). **Child: ** 48.7% female **Mean Age:** 11.93 (range 7–16 years) **BMI: ** 55.3% overweight 30.3% obesity **Ethnicity: ** 77.6% Hispanic	**Design:** Empirical, CS **Measures:** Self-reports **CG Attachment: ** Experiences in Close Relationship Structure (ECR-RS): anxiety and avoidance **Caregiving behavioral style:** Three Factor Eating Questionnaire (TFEQ-R18V2): hyper-activated and deactivated **Modeling of obesogenic behaviors: ** The Family Health Behavior Scale (FHBS) **CG and child obesogenic behaviors: ** The Caregiving System Scale (CSS): individual dietary intake; physical expenditure; health behaviors; self-regulation of eating; uncontrolled and emotional eating. **Child Attachment: ** Parent Modeling of Eating Behaviors Scale (PARM): anxiety and avoidance **Objective measure of the child’s BMI:** Measured by trained research team	**Insecure attached children:** - At greater risk of consuming high-calorie foods (*p* < 0.05). **Significant associations:** - Between secure avoidant attached children and obesogenic behaviors (*p* < 0.05). - Mediated by their eating self-regulation (*p* < 0.05). - Between the parent’s obesogenic behavior and the modeling of obesogenic behavior (*p* < 0.01). **No significant associations:** - Between the modeling of obesogenic behaviors, the reporting of parental obesogenic behaviors and the child’s report of obesogenic behaviors.

## Data Availability

Data is available in the included studies since it is a review.

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
