# Peer review of "Attachment, Feeding Practices, Family Routines and Childhood Obesity: A Systematic Review of the Literature"

_ijerph, 2023, doi:10.3390/ijerph20085496_

Round 1
Reviewer 1 Report
This study is a systematic analysis of this topic, which is of great research value, and we would like to acknowledge the researcher's commitment to this area. The following are some suggestions for the researcher to make this study more informative.
1. This study was analyzed through a systematic review of the literature, so the quality of each study is an essential reference base. It would be helpful if the authors further explained how the quality of the literature was subsequently evaluated after the deletion of irrelevant literature.
2.After L179, although the researcher listed all the literature based on age, there were only ten final papers, which is a small number, suggesting that the researcher can further explain whether the studies made under this circumstance have speculative characteristics. In addition, if the researcher can list the titles of the selected literature, it can be used as a reference for subsequent studies.
3. In 3.3. The papers' analysis section, although the authors have cited some literature to support their thesis to make the study have a more affluent idea, suggests that the authors can add more explanatory literature (L246-257) (L259-273)
4. The Discussion section hopes that the researcher can make an overall conclusion on the primary outcome, compare and discuss the results with other existing evidence in many aspects, and provide insight into the practical application of the results of this study so that this study can produce more academic contributions.
Reviewer 2 Report
· No need to divide the introduction into sections. Please delete Lines 44-45, 46, 67,101 and 117.
· Line 74-85: How do the authors differentiate between parenting practice and parenting style? Authoritative, authoritarian, permissive and disengaged are types of parenting style not practice. Parental warmth and control could be considered as parenting practice. Authors should expand on these types as a possible influencer on the changes in children's dietary intake. I would recommend referring to these articles (Psychol Health. 2020, 35(11): 1326-1345; Ecol Food Nutr. 2015, 54, 93-113).
· Line 148-150: Search terms should be clearly defined. Please use AND, and OR. For example, Child obesity AND parental control OR parental warmth AND….
· Reporting on quality assessment is seriously lacking in very important details. Please could the authors report who make the assessments, whether assessments conducted by two independent reviewers or by one reviewer and then cross-checked by another reviewer, and how disagreements would be dealt with?
· The risk of bias assessment is not reported. Bias assessments are important because they will help to determine how reliable the included studies are.
· Line 224-226: It is unclear how the quality was assessed (Good, fair, poor). A table that helps to understand each study is needed. Table A1 is not helpful.
· Tables 2-4: I would suggest adding significant P-values in key findings column.
· Several sentences/statements/terms throughout the paper should not be italicized.
· Ref # 55 is very old- Please delete.
· Line 459: Please follow the journal guideline for referencing.
Reviewer 3 Report
Thank you very much for having me to review the paper.
All the sections are correctly elaborated and justified. My congratulations to the authors, very good job.
The topic is original in that it addresses a current problem such as childhood obesity by incorporating behavioural psychological variables that may determine particular eating behaviour.
The objectives are, on the one hand, to synthesise data on the links between the quality of child attachment, parental feeding practices, family routines and childhood obesity risk; on the other hand, to assess whether these links are mediated by self-regulatory capacities. In addition, the paper addresses two gaps in the literature on the development of childhood obesity: 1) The analysis of multifactorial and transactional data to better understand children's dietary choices and weight trajectories during their development. 2) Analysis of this issue across different developmental stages to understand how childhood obesity risks may evolve.
Being able to analyse and determine behavioural data in the aetiology of childhood obesity is important to initiate a more appropriate therapy and even prevention approach.
I believe that the choice of these variables is fully justified in the literature.
The conclusions are consistent with the evidence and arguments presented.
The paper is very well written.
Author Response
Response to Reviewer 3 Comments :
Thank you very much for your comment. We are happy that our work interested you.
Cordially,
Sarah.
Round 2
Reviewer 2 Report
No further comments.